# School reopening and risks accelerating the COVID-19 pandemic: A systematic review and meta-analysis protocol

**Luís Carlos Lopes-Júnior** [ORCID]*[◐], **Priscila Carminati Siqueira, Ethel Leonor Noia Maciel**[◐]

Health Sciences Center at the Federal University of Espírito Santo (UFES), Vitoria, Espírito Santo, Brazil

◐ These authors contributed equally to this work.
* lopesjr.lc@gmail.com

**Funding:** The author(s) received no specific funding for this work.

**Competing interests:** The authors have declared that no competing interests exist.

## Abstract

### Background

One of the most recent concerns of this pandemic regards the role of schools reopening in disease transmission, as well as the impact of keeping schools closed. While school reopening seems critical for the education and mental health of children, adolescents, and adults, so far the literature has not systematically reached a consensus whether to recommend the return to schools in a way that would be safe for students and staff.

### Objective

To synthesize and critically evaluate the scientific evidence on the potential risk of accelerating the Coronavirus Disease 2019 (COVID-19) pandemic among children, adolescents, young adults, and adults with school reopening.

### Methods

This systematic review and meta-analysis protocol was elaborated following the PRISMA-P. We will include all observational study designs, which report on the potential risk of accelerating the COVID-2019 pandemic with school reopening. Electronic databases included were MEDLINE/PubMed, Cochrane Library, EMBASE, Web of Science, SCOPUS and CNKI. Additional sources will be also retrieved, including Clinical trials.gov-NIH, The British Library, Pro Quest Dissertations Database, Public Health Gray Literature Sources and Health Evidence, Google Scholar, and pre-prints [medRXiv]. No restriction to language or date will be used as search strategy. In an independently manner, two investigators will select studies, perform data extraction, as well as perform a critical appraisal of the risk of bias and overall quality of the selected observational studies, based on their designs. The heterogeneity among the studies will be assessed using the $I^2$ statistic test. According to the results of this test, we will verify whether a meta-analysis is feasible. If feasibility is confirmed, a random-effect model analysis will be carried out. For data analysis, the calculation of the pooled effect estimates will consider a 95% CI and alpha will be set in 0.05 using the R statistical software, v.4.0.4. In addition, we will rate the certainty of evidence based on

Cochrane methods and in accordance with the Grading of Recommendations Assessment, Development and Evaluation (GRADE).

## Expected results

This systematic review and meta-analysis will provide better insights into safety in the return to school in the context of the COVID-2019 pandemic, at a time when vaccination advances unevenly in several countries around the world. Hence, consistent data and robust evidence will be provided to help decision-makers and stakeholders in the current pandemic scenario.

## PROSPERO registration number

CRD42021265283; https://clinicaltrials.gov.

## 1. Introduction

Globally, a total of 192,284,207 confirmed cases of the Coronavirus Disease 2019 (COVID-19) caused by the new coronavirus (2019-nCoV), and 4,136,518 deaths have occurred to date, according to the report by the World Health Organization (WHO) on July 23, 2021. Additionally, a total of 3,568,861,733 vaccine doses have been administered so far [1].

It should be highlighted that in response to the 2019-nCoV, more than 100 countries had imposed national school closures by March 2020 to reduce the transmission rate [2]. Data available from 29 countries have shown that the proportion of children among COVID-19 cases varies from 0.3% to 13.8% [3]. Furthermore, 2019-nCoV infection in children is often asymptomatic, and most cases result in mild disease and better prognosis compared to adults [4–6]. However, data on the extent of the spread of 2019-nCoV among children/adolescents in the household setting, including transmission to elderly people, for example, who are at increased risk for severe disease, are still scarce [7,8].

An observational retrospective study comprising camp attendees and 526 household contacts, which assessed the secondary transmission and associated factors, has shown that the 2019-nCoV transmission from school-aged children/adolescents to household members led to the hospitalization of adults with secondary cases of COVID-19. In addition, considering the households where transmission occurred, 50% of the household contacts were infected [9].

Although many schools were closed by March 2020, several countries worldwide have now reopened schools for in-person teaching [10]. In fact, insufficient evidence is available to support school reopening in what regards the COVID-19 transmission rate and the harms to children/adolescents [11]. A study comprising 12,000,000,000 adults in the United Kingdom (UK) has found no statistical significance regarding the risk of death from the COVID-19 in households with or without children [12,13].

Additionally, a recent infection survey using molecular tests via polymerase chain reaction [14,15] showed that around 0.5–1% of children have a positive result. Also, earlier studies, including data from five countries [16–20], where all individuals were tested regardless of symptoms, found that transmission rates were considered low, mainly among elementary school children [16–20].

Data from ecological studies showed an association between in-person teaching and the speed as well as the extent of the 2019-nCoV community transmission [21–23], although these findings were not uniform across studies [24]. Despite the occurrence of several outbreaks in

schools [25–27], studies of outbreaks in places other than schools and school-like settings have pointed out that, when mitigation measures were in effect, transmission within schools was considered low; hence, infection rates represent that of the surrounding community [28,29].

Indeed, one of the most recent concerns of this pandemic is the role of schools in the transmission, as well as the impact of keeping schools closed. The exact risk for 2019-nCoV infection posed to individuals and household contacts, as well as communities, by in-person teaching in the schools is still an ongoing heated debate. While a safe reopening of schools seems feasible if mitigation measures are followed, there is a lack of robust evidence to support this return to school [30]. Modelling data undertaken by UK universities suggest that at least 30,000 more deaths from COVID-19 are estimated due to the reopening of school settings [31].

Evidence suggests that younger children (<10 y.o.) are less susceptible to infection when exposed to the virus [32]. Nonetheless, it has not been well established yet whether they are less likely to transmit the virus once infected [33,34] or whether this low susceptibility is compensated by the increased number of face-to-face contacts at school [35]. Recent evidence from a survey held in the United State (US) has demonstrated an increased risk of COVID-19 in what regarding outcomes among respondents living with a child who attends in-person teaching in school settings [30].

A recent living meta-analysis that investigated the extent of 2019-nCoV transmission in in-person teaching pointed out that overall infection attack rate and 2019-nCoV positivity rate in the school settings are low. Particularly, lower infection attack rate (IAR) and positivity rates were reported in students compared to school staff [5].

While school reopening seems pivotal to the education and mental health of children/adolescents/adults, so far, the literature has not systematically agreed on whether to recommend the return of students and staff to school in a safe manner [31]. To fully reopening schools in the setting of high community transmission, especially in view of the circulation of new variants of the new coronavirus, such as the delta variant, and with the very uneven speed of vaccination in the countries, it is essential to gather and synthesize the evidence-based studies to analyze this intervention in the pandemic scenario carefully. In this regard, gathering evidence on how 2019-nCoV transmission occurs in in-person teaching could support decision making on appropriate and safe school closure or school reopening based on robust evidence. In this study, we will aim to synthesize and evaluate the evidence on the potential risk of accelerating the COVID-19 pandemic in children, adolescents, young adults, and adults with school reopening.

## 2. Methods

This protocol has been elaborated following PRISMA-P [36]. Additionally, the protocol was registered with PROSPERO/UK (registration ID: CRD42021265283). The systematic review and meta-analysis will also be reported based on MOOSE guidelines [37] added to the PRISMA 2020 statement [38].

### Search strategy

The search strategy will be performed in six electronic bibliographic databases: MEDLINE/PubMed (from 1947 to August 30, 2021), EMBASE (Excerpta Medica dataBASE) (from 1973 to August 30, 2021), Cochrane Library (from 1991 to August 30, 2021), Web of Science (from 1985 to August 30, 2021), SCOPUS (from 2004 to August 30, 2021), and CNKI (from 1996 to August 30, 2021). Additional sources will be also searched, including Clinical trials.gov-NIH, The British Library, Pro Quest Dissertations Database, Public Health Gray Literature, Google

**Table 1. PECO acronym for search strategy.**

| PECOS component [40] | Inclusion criteria | Exclusion criteria |
|---|---|---|
| P–Population | Infant, Child, Child, Preschool, Adolescents, Young Adult, Adult (according to MeSH terms) of both sexes, of any ethnicity | Elderly people |
| E–Exposure | School reopening | – |
| C–Comparison | School lockdown | – |
| O–Outcome | Primary endpoint: risks accelerating the Coronavirus Disease 2019 pandemic<br>Secondary endpoints: viral load among children and teachers and transmission rate | |
| S–Study Design | Observational studies | Qualitative studies |

Scholar, and pre-prints for Health Sciences [medRXiv]. No restriction to language or date will be employed in the search strategy. Furthermore, we will also scrutinize the reference lists of articles searched to seek additional studies [39]. The PECOS (Population, Exposure/Comparison/Outcomes/Study Design) acronym [40] was used to elaborate our research question: "*Which evidence is available from observational studies on school reopening, and does this have an impact on the transmission rate of* Coronavirus Disease 2019 *pandemic among children, adolescents, young adults, and adults*?*"*, as depicted in Table 1.

Moreover, the EndNote™ will be used to organize as well as manage all the studies retrieved. Study selection will be carried out by two independent researchers (LCLJ and ELNM) using the Rayyan™ application, as an auxiliary tool for archiving, organizing, and selecting articles. First, we will screen for controlled descriptors, for instance, MeSH terms, Emtree terms, and their synonyms, as well as keywords will be identified. The Boolean operators "AND" and "OR" will be employed to combine the descriptors [39,41,42]. The preliminary pilot search strategy combining MeSH terms, synonyms (entry terms) as well as keywords that will be used in MEDLINE/PubMed is shown in Table 2.

## Eligibility criteria

In this study, we will include all observational study designs, i.e., cross-sectional, cohort, case-control, and ecological studies. Handsearching will be carried out in the reference lists and in the gray literature seeking additional studies. Moreover, no language and date restrictions will be employed in the search strategy.

## Study selection

First, all the records scrutinized from the six databases will be imported in the EndNote™. Afterwards, the duplicate articles will be removed. Two independent researchers (LCLJ and PCS) will search and screen the article records by titles and abstract using the Rayyan™ application. After the initial screening, the full text of the studies retrieved will be assessed for inclusion/exclusion by two independent researchers using Rayyan™ application. Disagreements in selected studies will be resolved by a third reviewer (ELNM). A flowchart will summarize the study selection process in line with the PRISMA 2020 statement [38] (Fig 1).

## Data extraction

Reviews or qualitative studies will be excluded in this systematic review. In addition, primary studies that did not meet the inclusion criteria will be excluded according to the reasons for

**Table 2. Preliminary pilot search strategy in MEDLLINE/PubMed.**

| Database | Search strategy |
|---|---|
| MEDLINE/ PubMed | **#1** ((Infant [MeSH Terms] OR Infants [All Fields] OR Child, Preschool [MeSH Terms] OR Preschool Child [All Fields] OR Children, Preschool [All Fields] OR Preschool Children [All Fields] OR Child [MeSH Terms] OR Children [All Fields] OR Adolescent [MeSH Terms] OR Adolescents [All Fields] OR Teens [All Fields] OR Teenagers [All Fields] OR Teenager [All Fields] OR Youth [All Fields] OR Youths [All Fields] OR Young Adult [MeSH Terms] OR Adult [MeSH Terms] OR Adult [All Fields])) |
| | **#2** ((Schools [MeSH Terms] OR School [All Fields] OR Primary School [All Fields] OR Secondary School [All Fields] OR High School [All Fields] OR School Teachers [MeSH Terms] OR High School Teachers [All Fields] OR Elementary School Teachers [All Fields] OR Pre-School Teachers [All Fields] OR Education, Graduate [MeSH Terms] OR Graduate Education [All Fields] OR Educations, Graduate [All Fields] OR Students [MeSH Terms] OR Student [All Fields] OR School Enrollment [All Fields] OR Enrollment, School [All Fields] OR School Enrollments [All Fields] OR Return to School [MeSH Terms] OR Return to Schools [All Fields] OR School, Return to [All Fields] OR School Reopening [All Fields])) |
| | **#3:** #1 AND #2 |
| | **#4** ((COVID-19 [MeSH Terms] OR COVID-19 Virus Disease [All Fields] OR COVID-19 Virus Infection [All Fields] OR 2019-nCoV Infection [All Fields] OR Coronavirus Disease-19 [All Fields] OR 2019 Novel Coronavirus Disease [All Fields] OR 2019 Novel Coronavirus Infection [All Fields] OR SARS Coronavirus 2 Infection [All Fields] OR SARS-CoV-2 Infection [All Fields] OR COVID-19 Pandemic [All Fields] OR Pandemic, COVID-19 OR SARS-CoV-2 [MeSH Terms] OR Coronavirus Disease 2019 Virus [All Fields] OR 2019 Novel Coronavirus [All Fields] OR SARS-CoV-2 Virus [All Fields] OR 2019-nCoV [All Fields] OR Wuhan Coronavirus [All Fields] OR SARS Coronavirus 2 [All Fields] OR Severe Acute Respiratory Syndrome Coronavirus 2 [All Fields] OR SARS-CoV-2 variants [Supplementary Concept] OR COVID-19 Virus variants [All Fields])) |
| | **#5:** #3 AND #4 |

PRISMA 2020 flow diagram for new systematic reviews which included searches of databases, registers and other sources

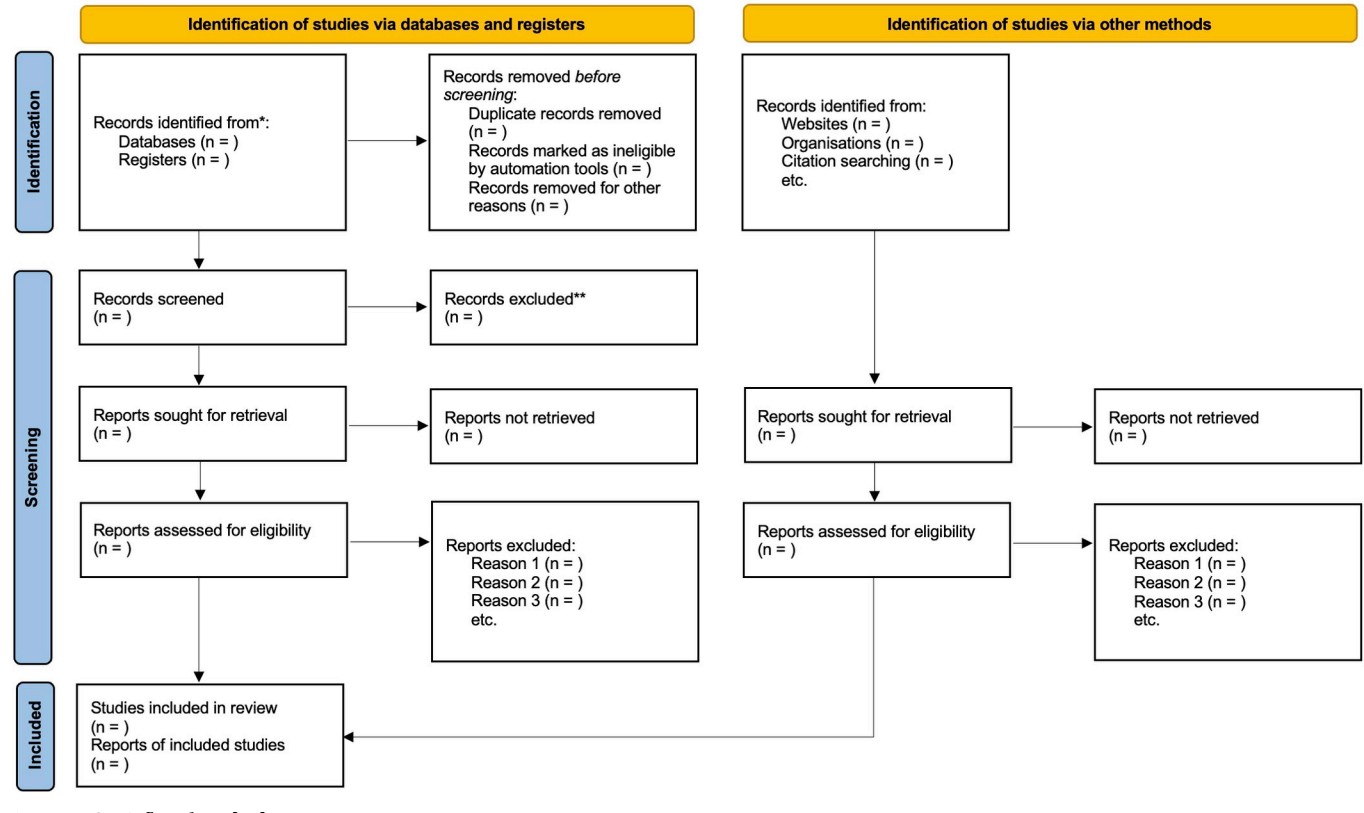

**Fig 1. PRISMA flowchart [38].**

exclusion. The two review authors (LCLJ and PCS) will perform data extraction for each included study based on forms published previously [39,41–44]. The expected completion date for this systematic review is September 30, 2021, due to the urgency to bring this evidence to light, in the current scenario in which the delta variant is spreading rapidly in several countries.

The same two reviewers (LCLJ and PCS) will perform data extraction independently. Information to be extracted includes: a) identification of the study and objectives; b) study population and baseline characteristics; c) type of exposure; d) study methodology; e) recruitment methods; f) times of measurement; g) follow-up; h) outcomes; i) main findings; j) clinical and epidemiological significance; and k) conclusions. S1 File shows the standardized form created for data extraction.

## Critical appraisal of included studies

Since this systematic review will include observational studies of different designs, the quality assessment tools should be appropriate and specific for each design. In this respect, Newcastle-Ottawa Scale (NOS) [45] will be used for evaluating the internal validity and risk of bias for cohort studies. For case-control studies, The Critical Appraisal Skills Programme (CASP) tool [46] will be employed. Regarding the cross-sectional studies, we will assess them using The Agency for Healthcare Research and Quality (AHRQ) tool [47]. In addition, the Methodological Index for Non-Randomized Studies-MINORS will be used for other study designs [48]. Two reviewers (LCLJ and ELNM) will perform the methodological assessment independently.

## Data synthesis

Study characteristics will be summarized and presented in tables. Heterogeneity among studies will be measured by the $I^2$ statistic to estimate the percentage of variation across studies, ranging from 0% to 100% [49,50], and its interpretation is: a) $I^2$ = 0%–40% indicates low heterogeneity; b) $I^2$ = 30%–60% shows moderate heterogeneity; c) $I^2$ = 50%–90% represents a substantial heterogeneity; and d) $I^2$ = 75%–100% indicates high heterogeneity [50,51].

Moreover, the subgroup analysis will be performed using a a random-effect model analysis adjusted for age, ethnicity, and sex, level of education, and non-pharmacological measures to mitigate the spread of the virus. In addition, will be carried out to explore the heterogeneity across studies. According to the $I^2$ statistic, we will determine whether a meta-analysis is feasible [43,44,51]. For data analysis, the calculation of the pooled effect estimates will consider a 95% CI and alpha set in 0.05 using the R statistical software v. 4.0.4. Besides, we will rate the certainty of evidence based on Cochrane methods and in accordance with the Grading of Recommendations Assessment, Development and Evaluation (GRADE) [52]. The evaluation of the quality of evidence in the evaluated studies will be independently performed in a paired manner by 2 reviewers (LCLJ and PCS). Disagreements will be addressed by a third reviewer (ELNM).

## Ethical aspects and dissemination plans

Because this is a systematic review protocol, and since our analysis will only include previously published data, the Institutional Review Board approval was not applicable. Moreover, the systematic review and meta-analysis will be reported following the PRISMA 2020 statement. Additionally, any amendments to this protocol will be registered. Regarding the dissemination plans, we intend to disseminate the results via peer-reviewed publication and via presentations in international conferences.

## 3. Limitations and strengths of this study

Two main potential limitations include the predominance of cross-sectional studies that might limit the generalizability of the results, mainly with regards to direction of cause-effect, given the study design; and the risk of type-1 error (arising from the bias in participants recruiting, confounding factors, subgroup, and sensitivity analyses of the potential studies).

However, the systematic review will provide an up-to-date synthesis of the estimation of potential risk of school reopening to accelerate the COVID-19 pandemic. Hence, a meta-analysis with reliable estimates of the prevalence rate and potential risk factors for the return to school will be provided. Finally, the results of this systematic review and meta-analysis will address this gap in the literature and support decision makers with evidence-based data to be considered in this complex context.

## 4. Conclusion

This systematic review and meta-analysis, to the best of our knowledge, will be the first to critically evaluate the scientific evidence and estimate the potential risk of school reopening to accelerate the COVID-19 pandemic among children, adolescents, young adults, and adults. Hence, we will provide better insights into safety in the return to school in the pandemic context, at a time when vaccination advances unevenly in several countries worldwide. Finally, we believe that this study will provide consistent evidence that will aid the decision makers and stakeholders in the current pandemic scenario.

## Supporting information

**S1 Checklist. PRISMA-P 2015 checklist.**
(DOCX)

**S1 File. Data extraction form based on previous publications [39,41–44].**
(DOCX)

## Author Contributions

**Conceptualization:** Luís Carlos Lopes-Júnior, Ethel Leonor Noia Maciel.

**Data curation:** Luís Carlos Lopes-Júnior, Ethel Leonor Noia Maciel.

**Formal analysis:** Luís Carlos Lopes-Júnior, Ethel Leonor Noia Maciel.

**Investigation:** Luís Carlos Lopes-Júnior, Priscila Carminati Siqueira.

**Methodology:** Luís Carlos Lopes-Júnior.

**Project administration:** Luís Carlos Lopes-Júnior.

**Supervision:** Luís Carlos Lopes-Júnior, Ethel Leonor Noia Maciel.

**Validation:** Luís Carlos Lopes-Júnior.

**Visualization:** Luís Carlos Lopes-Júnior, Priscila Carminati Siqueira.

**Writing – original draft:** Luís Carlos Lopes-Júnior, Priscila Carminati Siqueira, Ethel Leonor Noia Maciel.

**Writing – review & editing:** Luís Carlos Lopes-Júnior, Priscila Carminati Siqueira, Ethel Leonor Noia Maciel.

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
