## [Decision Letter · Decision Letter 0]

30 Sep 2021

PONE-D-21-25744School reopening and risks accelerating the COVID-19 pandemic: a systematic review and meta-analysis protocolPLOS ONE

Dear,

Thank you for submitting your manuscript to PLOS ONE. After careful consideration, we feel that it has merit but does not fully meet PLOS ONE’s publication criteria as it currently stands. Therefore, we invite you to submit a revised version of the manuscript that addresses the points raised during the review process. Please submit your revised manuscript by 15 October 2021. If you will need more time than this to complete your revisions, please reply to this message or contact the journal office at plosone@plos.org. Please include the following items when submitting your revised manuscript:A rebuttal letter that responds to each point raised by the academic editor and reviewer(s). You should upload this letter as a separate file labeled 'Response to Reviewers'.A marked-up copy of your manuscript that highlights changes made to the original version. You should upload this as a separate file labeled 'Revised Manuscript with Track Changes'.An unmarked version of your revised paper without tracked changes. You should upload this as a separate file labeled 'Manuscript'.If applicable, we recommend that you deposit your laboratory protocols in protocols.io to enhance the reproducibility of your results. Protocols.io assigns your protocol its own identifier (DOI) so that it can be cited independently in the future. For instructions see: https://journals.plos.org/plosone/s/submission-guidelines#loc-laboratory-protocols. Additionally, PLOS ONE offers an option for publishing peer-reviewed Lab Protocol articles, which describe protocols hosted on protocols.io. Read more information on sharing protocols at https://plos.org/protocols?utm_medium=editorial-email&utm_source=authorletters&utm_campaign=protocols.

We look forward to receiving your revised manuscript.

Kind regards,

Muhammad Shahzad Aslam, Ph.D.,M.Phil., Pharm-D

Academic Editor

PLOS ONE

Journal Requirements:

Reviewers' comments:

Reviewer's Responses to Questions

**Comments to the Author**

1. Does the manuscript provide a valid rationale for the proposed study, with clearly identified and justified research questions?

Reviewer #1: Yes

Reviewer #2: Yes

2. Is the protocol technically sound and planned in a manner that will lead to a meaningful outcome and allow testing the stated hypotheses?

Reviewer #1: Yes

Reviewer #2: Yes

3. Is the methodology feasible and described in sufficient detail to allow the work to be replicable?

Reviewer #1: Yes

Reviewer #2: Yes

4. Have the authors described where all data underlying the findings will be made available when the study is complete?

Reviewer #1: Yes

Reviewer #2: Yes

5. Is the manuscript presented in an intelligible fashion and written in standard English?

Reviewer #1: Yes

Reviewer #2: No

6. Review Comments to the Author

You may also provide optional suggestions and comments to authors that they might find helpful in planning their study.

Reviewer #1: Interesting and timely analysis

Some issues

I'm not sure that authors can use fixed effect with a meta-analysis of observational studies

Methods: school and pre school are different managed. For example face masks in some countries are compulsory for school but not pre school

Methods IF should not be included, actually it does not always reflect high quality paper

Reviewer #2: Congratulation to the authors. This is a great contribution to science, particularly in the COVID-19 pandemia. Some few suggestions and comments:

Review the standard english, particularly in methods section.

Put the extract form of information in the supplementary material;

In the manuscript, authors pointed that they will take into account the degree of vaccination. How this topic would be taken? Some explanation will be great.

Why authors will search all database without date restrictions, since the COVID-19 pandemic started in 2019?

The authors will take into account the mandatory or not use of mask in schools? This topic did not appear in the protocol.

7. PLOS authors have the option to publish the peer review history of their article (what does this mean?). If published, this will include your full peer review and any attached files.

Reviewer #1: **Yes: **Fabrizio D'Ascenzo

Reviewer #2: No

---

## [Author Response · Author response to Decision Letter 0]

12 Oct 2021

LETTER IN RESPONSE TO REVIEWERS

Vitória-ES, Brazil, October 2, 2021

Dear Editorial Board of the PLoS One

We are resubmitting our Manuscript ID: PONE-D-21-25744, entitled “School reopening and risks accelerating the COVID-19 pandemic: a systematic review and meta-analysis protocol”. We sincerely appreciate your valuable feedback and would like to thank you for all suggestions and comments. The editorial as well as the reviewers request was addressed and summarized below with changes in the main text. All co-authors have agreed to the resubmission after these revisions. 

Below we have responded in detail point by point the items raised by reviewers, which have all been addressed.

Muhammad Shahzad Aslam, Ph.D.,M.Phil., Pharm-D

Academic Editor - PLOS ONE

Journal Requirements:

Response: Ok. Done. 

Response: Ok. Done. 

Upon re-submitting your revised manuscript, please upload your study’s minimal underlying data set as either Supporting Information files or to a stable, public repository and include the relevant URLs, DOIs, or accession numbers within your revised cover letter. For a list of acceptable repositories, please seehttp://journals.plos.org/plosone/s/data-availability#loc-recommended-repositories. Any potentially identifying patient information must be fully anonymized.

Response: Ok. Done. 

Reviewers' comments:

Reviewer's Responses to Questions

Comments to the Author

1. Does the manuscript provide a valid rationale for the proposed study, with clearly identified and justified research questions?

Reviewer #1: Yes

Reviewer #2: Yes

2. Is the protocol technically sound and planned in a manner that will lead to a meaningful outcome and allow testing the stated hypotheses?

Reviewer #1: Yes

Reviewer #2: Yes

3. Is the methodology feasible and described in sufficient detail to allow the work to be replicable?

Reviewer #1: Yes

Reviewer #2: Yes

4. Have the authors described where all data underlying the findings will be made available when the study is complete?

Reviewer #1: Yes

Reviewer #2: Yes

5. Is the manuscript presented in an intelligible fashion and written in standard English?

Reviewer #1: Yes

Reviewer #2: No

6. Review Comments to the Author

You may also provide optional suggestions and comments to authors that they might find helpful in planning their study.

Reviewer #1:

Interesting and timely analysis.

Response: Thank you so much for your comments as well as suggestions.

Some issues

- I'm not sure that authors can use fixed effect with a meta-analysis of observational studies

Response: You are right. We made adjustments in the data synthesis and into abstract. Thank you very much.

In fact, random effects models that assume that the effect of interest is not the same across studies. In this sense, they consider that the studies that are part of the meta-analysis form a random sample of a hypothetical population of studies. For this reason, they create combined results with a greater confidence interval, which is why they are the most recommended. However, despite having this advantage, methods with random effects are criticized for giving greater weight to smaller studies (Egger et al., 2001).

Egger, M.; Smith, G.D. & Altman, D.G. (Ed.s) (2001). Systematic reviews in health care: Meta-analysis in context. 2nd ed. London: BMJ Books.

- Methods: school and pre school are different managed. For example face masks in some countries are compulsory for school but not pre school

Response: Thank you for raising this timely issue. Actually, the results of this systematic review also will be presented stratifying them by level of education. We have add this on data synthesis. Thanks!

Methods IF should not be included, actually it does not always reflect high quality paper

Response: OK. We agree with you. We have removed the IF from the data extraction form as per suggested.

Reviewer #2:

Congratulation to the authors. This is a great contribution to science, particularly in the COVID-19 pandemia. 

Response: Thank you so much for your comments as well as suggestions.

Some few suggestions and comments:

Review the standard english, particularly in methods section.

Response: Ok. Done. The new version of this manuscript was edited for proper English language, grammar, punctuation, spelling, and overall style by two of the highly qualified native English speaking, as recommended by you. (Please, see attached the revision certificate).

Put the extract form of information in the supplementary material;

Response: Ok. Done. 

In the manuscript, authors pointed that they will take into account the degree of vaccination. How this topic would be taken? Some explanation will be great.

Response: Thank you very much for this interesting question. We reflected on this issue and decided to withdraw this analysis, once, it will be difficult to obtain data from each study regarding vaccination (comprising the period of primary data collection considering this endpoint). Instead, we thought of addressing the protective factor "vaccination" in the discussion section of our systematic review.

Why authors will search all database without date restrictions, since the COVID-19 pandemic started in 2019?

Response: Because according to the Cochrane Collaboration Handbook (Higgins et al., 2021), dates should not be restricted in systematic reviews. Although, we will obviously only retrieve articles from the end of 2019 onwards.

Higgins JPT, Thomas J, Chandler J, Cumpston M, Li T, Page MJ, Welch VA (editors). Cochrane Handbook for Systematic Reviews of Interventions version 6.2 (updated February 2021). Cochrane, 2021. 

The authors will take into account the mandatory or not use of mask in schools? This topic did not appear in the protocol.

Response: Thank you for your valuable comments. We have added this issue in data synthesis. Yes, we will take into account whether or not masks are mandatory in schools, as well as other measures to mitigate the spread of the virus. Based on the outcomes of the studies included in the review, we will perform subgroup analyses.

The authors certify that we have no funding or conflicts of interest to disclose. The manuscript is consistent with the Guidelines for Authors of the PLOS One. Also, this is an original study that has not been published previously and is not under consideration for publication in another journal. 

Yours Sincerely,

The authors

---

## [Decision Letter · Decision Letter 1]

4 Nov 2021

School reopening and risks accelerating the COVID-19 pandemic: a systematic review and meta-analysis protocol

PONE-D-21-25744R1

Dear,

We’re pleased to inform you that your manuscript has been judged scientifically suitable for publication and will be formally accepted for publication once it meets all outstanding technical requirements.

Kind regards,

Muhammad Shahzad Aslam, Ph.D.,M.Phil., Pharm-D

Academic Editor

PLOS ONE

Additional Editor Comments (optional):

Reviewers' comments:

Reviewer's Responses to Questions

**Comments to the Author**

1. Does the manuscript provide a valid rationale for the proposed study, with clearly identified and justified research questions?

Reviewer #1: No

Reviewer #2: Yes

2. Is the protocol technically sound and planned in a manner that will lead to a meaningful outcome and allow testing the stated hypotheses?

Reviewer #1: Yes

Reviewer #2: Yes

3. Is the methodology feasible and described in sufficient detail to allow the work to be replicable?

Reviewer #1: Yes

Reviewer #2: Yes

4. Have the authors described where all data underlying the findings will be made available when the study is complete?

Reviewer #1: Yes

Reviewer #2: Yes

5. Is the manuscript presented in an intelligible fashion and written in standard English?

Reviewer #1: Yes

Reviewer #2: Yes

6. Review Comments to the Author

You may also provide optional suggestions and comments to authors that they might find helpful in planning their study.

Reviewer #1: I have no competing interests. We thanks the authors for the interesting paper which should be accepted.

Reviewer #2: The authors adjusted the manuscript considering all the reviewers' recommendations.

Allied to this, it is an extremely relevant and current theme. I recommend the publication

7. PLOS authors have the option to publish the peer review history of their article (what does this mean?). If published, this will include your full peer review and any attached files.

Reviewer #1: **Yes: **Fabrizio D'Ascenzo

Reviewer #2: No

---

## [Editor Report · Acceptance letter]

9 Nov 2021

PONE-D-21-25744R1 

School reopening and risks accelerating the COVID-19 pandemic: a systematic review and meta-analysis protocol 

Dear Dr. Lopes-Junior:

I'm pleased to inform you that your manuscript has been deemed suitable for publication in PLOS ONE. Congratulations! Your manuscript is now with our production department. 

Kind regards, 

on behalf of

Dr. Muhammad Shahzad Aslam 

Academic Editor

PLOS ONE